# Calcium and Redox Liaison: A Key Role of Selenoprotein N in Skeletal Muscle

**DOI:** 10.3390/cells10051116

**Published:** 2021-05-06

**Authors:** Ester Zito, Ana Ferreiro

**Affiliations:** 1Istituto di Ricerche Farmacologiche Mario Negri IRCCS, 20156 Milan, Italy; 2Basic and Translational Myology Laboratory, UMR8251 Université de Paris/CNRS, 75013 Paris, France; ana.b.ferreiro@gmail.com; 3AP-HP, Reference Centre for Neuromuscular Disorders, Institut of Myology, Pitié-Salpêtrière Hospital, 75013 Paris, France

**Keywords:** stress of the endoplasmic reticulum, UPR (unfolded protein response), SEPN1, SEPN1-related myopathy, SELENON, SELENON-related myopathy, multi-minicore disease, calcium handling, redox homeostasis

## Abstract

Selenoprotein N (SEPN1) is a type II glycoprotein of the endoplasmic reticulum (ER) that senses calcium levels to tune the activity of the sarcoplasmic reticulum calcium pump (SERCA pump) through a redox-mediated mechanism, modulating ER calcium homeostasis. In SEPN1-depleted muscles, altered ER calcium homeostasis triggers ER stress, which induces CHOP-mediated malfunction, altering excitation–contraction coupling. SEPN1 is localized in a region of the ER where the latter is in close contact with mitochondria, i.e., the mitochondria-associated membranes (MAM), which are important for calcium mobilization from the ER to mitochondria. Accordingly, SEPN1-depleted models have impairment of both ER and mitochondria calcium regulation and ATP production. SEPN1-related myopathy (SEPN1-RM) is an inherited congenital muscle disease due to SEPN1 loss of function, whose main histopathological features are minicores, i.e., areas of mitochondria depletion and sarcomere disorganization in muscle fibers. SEPN1-RM presents with weakness involving predominantly axial and diaphragmatic muscles. Since there is currently no disease-modifying drug to treat this myopathy, analysis of SEPN1 function in parallel with that of the muscle phenotype in SEPN1 loss of function models should help in understanding the pathogenic basis of the disease and possibly point to novel drugs for therapy. The present essay recapitulates the novel biological findings on SEPN1 and how these reconcile with the muscle and bioenergetics phenotype of SEPN1-related myopathy.

## 1. Introduction

SELENON-related myopathy, also known as SEPN1-related myopathy (SEPN1-RM), designates a rare inherited myopathy, previously classified as four different disorders, caused by homozygous or compound heterozygous mutations in the *SELENON*, also known as *SEPN1*, gene (OMIM#606210) which lead to SEPN1 loss of function. The muscle phenotype of SEPN1-RM typically presents in infancy with severe weakness of neck and trunk muscles and impairment of diaphragm strength, which, in the absence of assisted ventilation, eventually leads to respiratory failure and death. Limb muscles are less severely affected and, in general, ambulation is preserved but, in severe cases, limb muscle weakness, together with fatigue, may lead to loss of ambulation [1,2]. NADH-TR and SDH staining on muscle biopsies of SEPN1-RM patients shows small areas negative to the staining, i.e., of mitochondrial depletion, corresponding at the ultrastructural level to foci of sarcomere disorganization, which are known as minicores and are a hallmark of this disease.

SEPN1 function is still somewhat elusive, mainly because of the difficulty of obtaining a pure SEPN1 protein, which would be important to assess its activity in vitro and to grow crystals, so as to understand the spatial organization of the protein. However, significant progress has been made in the last few years. SEPN1 is one of 25 selenoproteins identified in humans. All selenoproteins contain selenium in the form of the nucleophilic amino acid selenocysteine (U), and, in all of them except Selenoprotein P, U is localized at the catalytic site [3]. SEPN1 is a type II transmembrane glycoprotein of the ER with a short tail of amino acids on the cytosolic side and the rest of the protein, including an EF-hand domain and a thioredoxin-like (CU) domain, on the ER side. The EF-hand domain in SEPN1 can sense calcium fluctuations in the lumen of the ER [4]. When calcium levels decrease, the thioredoxin-like domain of SEPN1, containing the highly nucleophilic selenocysteine, is possibly more exposed. This facilitates redox modulation of its partners, including the sarcoplasmic/endoplasmic reticulum calcium pump (SERCA), which is activated to pump calcium from the cytosol into the ER [4]. ER calcium impairment in SEPN1-depleted models likely gives rise to ER stress, which triggers muscle malfunction through the CHOP branch of the ER stress response [5].

Furthermore, SEPN1 is localized at the MAMs, a region where ER is in close contact with mitochondria. Thus, it regulates not only calcium levels in the ER but also in mitochondria, in fine tuning ATP and hence muscle bioenergetics [6,7].

This review focuses on the recent biological findings on SEPN1 that help to rationalize the muscle phenotype of SEPN1-RM and could be important for designing a disease-modifying drug. We first describe the features of SEPN1 myopathy and briefly summarize the main features of the link between Ca^2+^ and redox homeostasis in the ER, to then turn to the specific roles of SEPN1 in the regulation of the Ca^2+^–redox link in the ER lumen and in mitochondrial Ca^2+^ content, OXPHOS function and ATP production.

## 2. SEPN1-Related Myopathy

Histopathologically, SEPN1-RM shows large variability, but multi-minicores, which are localized areas of mitochondria depletion and sarcomere disorganization within the myofibers, are found in the majority of muscle biopsies from SEPN1-RM patients. Furthermore, multi-minicores are often associated with type 1 fiber predominance and hypotrophy, mild dystrophic features and occasionally with eosinophilic protein inclusions (Figure 1). Therefore, most patients had the phenotypical diagnoses of the congenital myopathy multi-minicore disease or of Rigid Spine muscular dystrophy (RSMD) and a minority was initially classified as fiber-type disproportion or Mallory body-like desmin-related myopathy [8]. Core myopathies are often caused by autosomal dominant or recessive mutations in the *RYR1* gene, encoding the skeletal muscle ryanodine receptor RYR1, a redox-sensitive Ca^2+^ channel of the sarcoplasmic reticulum (SR). RYR1 triggers Ca^2+^ efflux from this cellular compartment into the cytosol, leading to excitation–contraction coupling [9,10]. A lower percentage of the autosomal recessive forms of multi-minicore disease are due to mutations of the Selenoprotein N gene (*SEPN1/SELENON*) [1,11]. Different types of mutation are associated with SEPN1-related myopathy, including missense variants, small duplications/insertions, deletions, nonsense and splice mutations and also CNVs (Copy Number Variations) [12,13,14]. Recently, *SEPN1* exon 1 was identified as the main mutational hotspot, and the first genotype–phenotype correlations were established, with bi-allelic null mutations significantly associated with higher disease severity [13].

In patients with *SEPN1* mutations, weakness is more severe in axial (neck and trunk) muscles and is often associated with diaphragm weakness and fatigue, leading to major scoliosis and potentially lethal respiratory failure, which are phenotypical hallmarks of SEPN1-RM. Mutations in *SEPN1* lead to loss of ambulation in 10% of cases and systematic functional decline from the end of the third decade [13,15]. *SEPN1* mutations are also associated with insulin resistance and abnormal oral glucose tolerance test (OGTT) [16]. No correlation has been found so far between abnormal glucose metabolism and the patients’ age or the type of *SEPN1* mutation. However, insulin resistance was observed, paradoxically, only in patients with an extremely low body mass index (BMI), suggesting that insulin resistance is part of the SEPN1-RM phenotype in these subjects [7].

Interestingly, a significant correlation between body weight and disease severity has been identified in SEPN1-RM patients. Underweight patients tend to have a moderate form of disease with preserved ambulation in adulthood, whereas the rare overweight patients reported had severe weakness of both weight-bearing and non-weight-bearing muscles, needed assisted ventilation early in childhood and lost ambulation in the second decade, confirming that alterations in BMI correlate with the phenotype of SEPN1-RM [6,13].

## 3. Redox Modulation of Calcium Handling in the ER and Mitochondria

The endoplasmic reticulum (ER) (and its specialized cognate in skeletal muscle, the sarcoplasmic reticulum SR) is a dynamic reservoir of calcium, where this ion is highly concentrated (0.1–1 mM) compared to the low nanomolar concentration in the cytosol [9,17] (Figure 2). The SR dynamically responds to electrical stimuli during the mechanism of excitation–contraction coupling by releasing and taking up calcium, thus favoring rapid calcium-induced muscle contraction and relaxation. Calcium uptake in the ER/SR is mediated by calcium pumps (sarco-endoplasmic reticulum calcium ATPase, SERCAs), whereas calcium release from ER/SR is mediated by the calcium channels Inositol-triphosphate receptor, IP3Rs, and Ryanodine receptors, RYRs.

The ER is in close contact with mitochondria in a 10–25-nm-wide region of juxtaposed membranes which hold together the two organelles by protein tethers and is referred to as mitochondria-associated membranes (MAMs) [19]. In some cases, ER tethers involve IP3R and RYR, suggesting an important role of the MAMs in calcium mobilization between ER and mitochondria [20,21,22]. In skeletal muscles, a quarter of the mitochondrial surface is in contact with the Calcium Release Unit (CRU) of the junctional SR through the MAMs, supporting the concept of high Ca^2+^ microdomains close to the CRU which lead to Ca^2+^ uptake by mitochondria [9,23,24]. Thus, the activity of calcium pumps and channels in the ER influences calcium content not only in the ER lumen but also in mitochondria.

Besides specific inhibitors (such as phospholamban for SERCA and calstabin for RYR), the activity of these calcium pumps and channels is tuned by chaperones (e.g., calnexin and calreticulin) and redox-active proteins (e.g., ERP57, ERP44, ERO1) [25,26,27,28,29,30]. For example, regarding the redox modulation of calcium, it is known that ERP57 oxidizes the two cysteines in the L4 luminal loop of SERCA2b, thus promoting a disulfide bond and leading to a decrease in SERCA activity, whereas the reduction of this disulfide bridge restores Ca^2+^ pump activity [25].

## 4. SEPN1-Mediated Calcium Handling in the ER and Mitochondria

SEPN1 is one of the 25 selenocysteine-containing selenoproteins and also the first that was associated with a genetic disease [3]. SEPN1 is ubiquitously expressed throughout the body and detected early in muscle precursors [31,32]. This type II glycoprotein of the ER has a short tail of amino acids exposed on the cytosolic side and the rest of the protein in the ER lumen. On the luminal side of the protein, there are two known domains: an EF-hand calcium-binding domain and a thioredoxin-like domain (CU) encompassing the U residue [31]. Biophysical data from our laboratories suggest that SEPN1 binds calcium through its EF-hand with an affinity constant in the concentration range of ER luminal calcium (130 < kd < 240 microM of a SEPN1 synthetic peptide including the EF-hand) and undergoes marked conformational changes upon calcium binding. When calcium is low in the ER, the thioredoxin-like domain of SEPN1 is more exposed and redox-modulates its partners by relaying to them electrons through the nucleophilic U, so the ER redox poise is more reduced [4]. Relevantly, the ATP-dependent calcium pump SERCA is one of the redox partners of SEPN1. Lack of SEPN1 reduces calcium uptake into the ER of cultured cells [33,34] and increases the relaxation time after electrical stimulation in flexor digitorum brevis (FDB) muscle fibers [5], indicating reduced calcium entry into the sarcoplasmic reticulum. Thus, considering that SERCA2 is activated after the reduction of two luminal cysteines in the L4 domain, SEPN1 is an intermediary between ER calcium handling and redox regulation that senses ER calcium levels and, through redox activity, triggers the SERCA-mediated refilling of the ER/SR calcium store in skeletal muscle [4,25,35] (Figure 2).

We have previously demonstrated a functional interaction regarding calcium handling between SEPN1 and the H_2_O_2_-generating protein disulfide oxidase endoplasmic oxidoreductin 1 (ERO1) [36]. The latter participates (together with PDI) in the relay of electrons for introducing disulfide bonds in proteins while generating H2O2 (Figure 3) [36]. Patient-derived and murine cells without SEPN1 show signs of oxidative stress while SEPN1 KO mouse muscles are hypersensitive to the lack of antioxidant vitamins and ERO1 overexpression, leading to reduced muscle force [8,33,34,37,38,39].

The expression of a hyperactive ERO1, which carries a mutation in one of the regulatory cysteines (Cy131), and thus locally generates more H_2_O_2_, acts cooperatively with SEPN1 loss in inhibiting SERCA activity. This suggests that when ERO1 is hyperactive, and in cells lacking SEPN1, SERCA is less active, supporting a scenario in which both ERO1 and SEPN1 regulate SERCA but in opposite ways: the former inactivates SERCA and the latter activates it. Thus, SEPN1 protects SERCA from ERO1-mediated hyperoxidation [5]. Early studies suggested that SEPN1 also works as a modifier of the RYR1 channel. However, these results could not be confirmed by later studies [5,40].

## 5. SEPN1 at ER–Mitochondria Contact Sites

SEPN1 is localized in a region of the ER referred to as MAMs, where this organelle is in contact with mitochondria and where ER-to-mitochondria calcium transfer occurs [6].

In light of this specific ER localization of SEPN1, it has been investigated whether SEPN1 defects impair MAMs and influence mitochondrial calcium handling. A shrinkage of the ER region in contact with mitochondria, i.e., less extended MAM, has been identified in SEPN1-depleted cells. However, at the moment, we do not know whether these alterations of MAMs in SEPN1-deficient models are due to a structural function of SEPN1, which would hold together ER and mitochondria, or are a consequence of other SEPN1-dependent alterations such as ER stress.

Consistently, in muscle biopsies of patients with *SEPN1* mutations leading to protein absence, mitochondria were misplaced from their triadic position, i.e., not on both sides of the CRU, leaving regions of muscle fibers where CRU were not coupled with mitochondria. This effect of SEPN1 loss on MAMs was coupled with impairment of mitochondrial calcium levels, suggesting that SEPN1 modulates contacts between the ER and mitochondria and calcium levels in both [6].

To meet metabolic demand during excitation–contraction coupling, skeletal muscle fibers need ATP, which is supplied by mitochondria through the OXPHOS system [41].

Since mitochondrial calcium regulates muscle OXPHOS and thus ATP production [42], it has also been explored whether SEPN1 loss affects the different complexes of the OXPHOS. SEPN1-devoid cells had a 25 to 30% lower ATP content with reduced complex I activity and a significant defect of the OXPHOS complexes II and IV. In line with this, isolated muscle fiber bundles from *Sepn1* knock-out (KO) mice showed a reduction of mitochondrial respiration that was more marked in slow and mixed-fiber muscles, suggesting that SEPN1 regulates OXPHOS and ATP production in skeletal muscle [6].

## 6. SEPN1 Loss and ER Stress

SEPN1 expression is induced following ER stress, a condition elicited when the load of unfolded proteins inside the ER exceeds the capacity of the folding machinery [33]. ER stress activates a series of corrective measures, defined as the Unfolded Protein Response (UPR), that, by activating three sensors (IRE1, PERK and ATF6), triggers induction of chaperones, attenuation of protein translation and boosting of protein degradation, to restore ER homeostasis [43]. Despite not being a highly secretory tissue, skeletal muscle is still sensitive to the triggers of ER stress, such as unbalanced calcium and hypoxia, and these elicit UPR in this tissue [44].

Unmitigated ER stress in skeletal muscle can also shift the UPR from adaptive to maladaptive by inducing the transcription of the mediator C/EBP homologous protein (CHOP), leading to reduced exercise tolerance and detrimental effects on skeletal muscle, suggesting that induction of CHOP in skeletal muscle during the UPR impairs muscle homeostasis after exercise [45]. In the UPR signaling, CHOP is upstream of ERO1, which acts as a mediator of the adaptive UPR given its role in oxidative protein folding but, on the other hand, is also a potential detrimental mediator given its ability to generate one molecule of the dangerous oxidant H2O2 for each disulfide bond introduced into client proteins [46,47].

CHOP and ERO1 were both upregulated in the diaphragm of *Sepn1* KO mice, which also had impaired strength, consistent with the diaphragmatic dysfunction of SEPN1-RM patients, and altered calcium handling in the FDB muscle [5,48]. CHOP deletion in *Sepn1* KO mice attenuated ERO1, rescued the diaphragmatic force and calcium handling and reduced the signs of ER stress, suggesting that the CHOP branch of the UPR impairs muscle force and calcium handling in *Sepn1* KO muscles. This also suggests that SEPN1, through its reductase activity, protects muscle cells from the hyperoxidation due to UPR-induced ERO1. Therefore, the CHOP/ERO1 pathway of the PERK branch of UPR is an important pathogenic component of SEPN1-related myopathy (Figure 3) [5,38].

The connection between lack of SEPN1 and insulin resistance in SEPN1-RM in relation to ER stress and the consequent maladaptive response has been explored. Skeletal muscle is the main contributor to post-prandial glucose uptake in the body, so defects in muscle-mediated glucose uptake reduce glucose tolerance and induce insulin resistance [49]. Insulin resistance is triggered by saturated fatty acid overload, which also induces ER stress and the consequent UPR [50]. Furthermore, alterations of MAM integrity in skeletal muscle have been associated with defective insulin sensitivity [51]. Given the ER stress-mediated regulation of SEPN1, the insulin-resistant phenotype of SEPN1-RM patients and their altered MAM integrity in skeletal muscle, studies have explored whether the lack of SEPN1 creates sensitization to ER stress induced by saturated fatty acid overload and the consequent maladaptive UPR, eliciting insulin resistance [6,7].

Interestingly, a lack of SEPN1 in saturated fatty acid-challenged myotubes does trigger ER stress, as indicated by the induction of CHOP, ERO1, GADD34 and XBP-1; it also disrupts ER-to-mitochondria communication and alters mitochondrial quality, together with cell bioenergetics. However, ubiquitous CHOP ablation only temporarily rescued the glucose intolerance of high-fat-fed *Sepn1* KO mice, suggesting that ER stress and the consequent maladaptive CHOP-mediated response is involved in the insulin resistance phenotype of *Sepn1* KO mice but also that the transient beneficial effect from constitutive CHOP ablation might be due to the fact that it leads to increased adiposity [52]. These findings suggest that environmental cues, such as a high-fat diet, which trigger ER stress in skeletal muscle, could modulate the pathological phenotype of SEPN1-related myopathy, contributing to the course of the disease beyond the genotype–phenotype correlation.

From the point of view of the correlation between ER stress and the muscle phenotype of the core disease, knock-in mice carrying the I4895T heterozygous mutation in the ER calcium channel RYR1, which impairs RYR1 activity without leading to calcium leak, display ER stress in skeletal muscle together with muscle dysfunction. Treatment with the chemical chaperone 4PBA reduces ER stress and improves muscle function. Although SEPN1 regulates SERCA and thus ER calcium uptake, whilst RYR1 regulates calcium release from ER, both SEPN1 and RYR1 mutants show ER calcium defects, which might represent a common underlying mechanism of ER stress and the pathological phenotype. This suggests that, similarly to *RYR1*^I4895T^, defects in calcium handling in the ER trigger ER stress and muscle impairment in SEPN1-RM, which can be rescued by a chemical chaperone that attenuates the stress [53].

## 7. Conclusion and Therapeutic Perspectives of Core Diseases

SEPN1 is a ubiquitously expressed type II glycoprotein of the ER that defends ER from hyperoxidation and calcium impairment through a redox-regulated mechanism that activates SERCA pumps. Accordingly, calcium impairment in SEPN1-deficient models hampers excitation–contraction coupling in muscle and triggers ER stress. Moreover, SEPN1 is localized at MAMs, a region of the ER in contact with mitochondria, which is important for calcium mobilization from the ER to mitochondria. Accordingly, SEPN1-deficient models present alterations of MAMs, mislocalization of mitochondria and impairment of mitochondrial calcium and ATP, a cause of muscle weakness.

There is a significant degree of clinical and histopathological overlap between RYR1- and SEPN1-RM, which might reflect common pathophysiological mechanisms.

Increases in intracellular oxidant activity and markers of oxidative stress were in fact found in primary cultured myotubes from patients with either *SEPN1* null or *RYR1* mutations. The antioxidant N-acetylcysteine (NAC) rescued the SEPN1- and the RYR1-associated cell phenotypes, enhancing the survival of patient-derived cells while reducing the hyperoxidized state of their proteins [54].

These results provided the experimental basis for the first therapeutic trials using NAC in SEPN1-RM and RYR1-RM (ClinicalTrials.gov NCT02505087 and NCT02362425). The outcome of the small SEPN1-RM pilot trial (SELNAC) is currently being analyzed. Unfortunately, the RYR1-RM trial with NAC showed only some muscle improvement but failed to achieve its primary endpoint [55]. Thus, there is a critical need for studies on the pathogenesis to identify other drugs synergistic with NAC, or alternative therapeutic options for SEPN1-RM and RYR1-RM.

ER stress and the consequent maladaptive UPR is emerging as a common trigger of SEPN1- and RYR1-RM, giving rise not only to common histopathological signs in the two genetically distinct myopathies but also to an overlapping pathological phenotype. As ER stress and the consequent UPR are druggable [56], it is conceivable that, in the near future, inhibitors of ER stress and its response could represent novel therapeutic options to be tested in clinical trials of SEPN1-RM and RYR1-RM.

## Figures and Tables

**Figure 1 cells-10-01116-f001:**
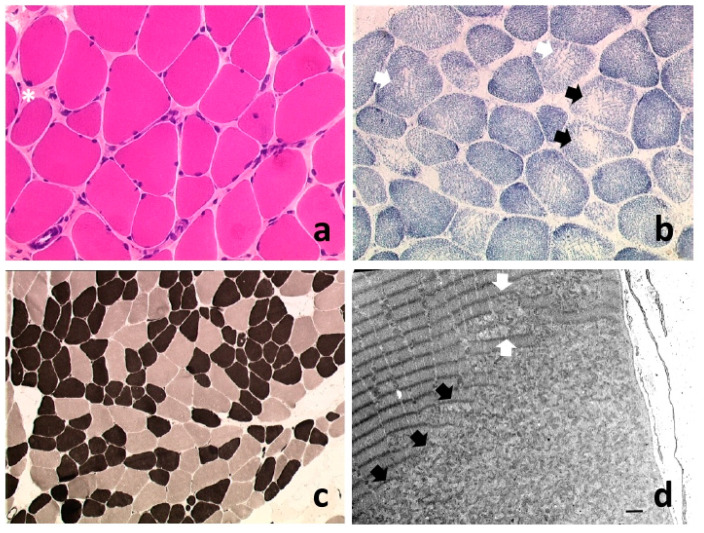
Main histopathological findings in SEPN1-RM. Transversal sections of frozen muscle samples (a–b 20×, c 10×); longitudinal electron microscopy section (d, bar = 2 µm). (**a**) Variability of muscle fiber diameter, with some rounded, smaller fibers surrounded by increased endomysial connective tissue (asterisk), while mild endomysial fibrosis is frequent, necrotic or regenerating fibers are rarely observed. (**b**) SDH showing fibers with cores (rounded areas devoid of oxidative (mitochondrial) activity, black arrows) or with uneven oxidative staining (white arrows). (**c**) Type 1 fiber predominance and hypotrophy: oxidative fibers (dark) are more abundant and smaller in diameter compared to type 2 glycolitic fibers (light colour). (**d**) Small (white arrows) and longer (black arrows) minicores, contrasting with normally organized sarcomeres and SR/mitochondrial complexes in adjacent areas of the muscle fiber (top left). Minicores are areas of mitochondria depletion and sarcomere disorganization which span a variable but limited number of sarcomeres along the fiber longitudinal axis. Hematoxylin–eosin (H&E) (**a**), SDH (**b**), Myosin ATPase pH 4.65 (**c**).

**Figure 2 cells-10-01116-f002:**
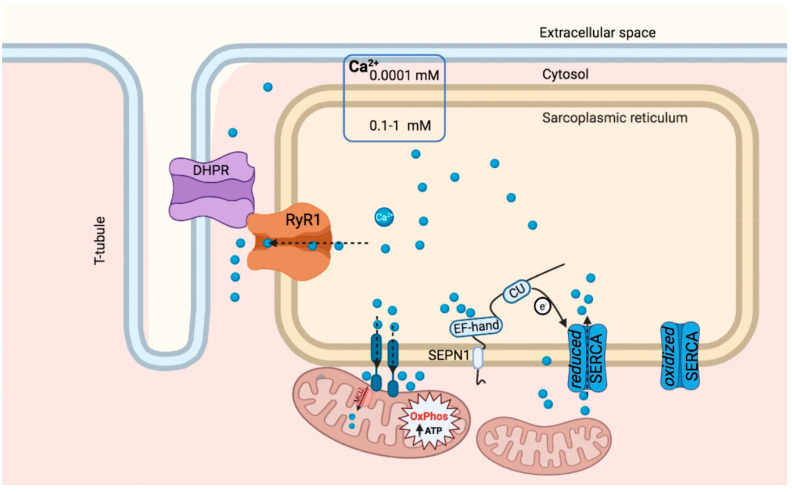
SEPN1-mediated calcium handling and muscle phenotype of SEPN1-RM. The functional relationship between SEPN1 and RYR1 in the SR is illustrated. Both proteins tune the ER calcium handling: RYR1 is a Ca^2+^ (blue spheres) release channel of the SR, which is activated by the dihydropyridine receptor (DHPR) on the T tubule, and SEPN1 senses ER calcium (blue spheres) levels through an EF-hand domain. Then, SEPN1 activates, through a redox reaction involving the thioredoxin-like domain CU (the arrow indicates an electron flow from the nucleophilic U), the calcium pump SERCA, which leads to calcium uptake into the ER. Ca^2+^ direction is indicated by dashed arrows. The range of intraluminal and cytosolic Ca^2+^ is indicated. SEPN1 is enriched at MAMs, a region of the ER in close contact with mitochondria through proteinaceous tethers, which is important for calcium mobilization from the ER to mitochondria. Calcium is released from ER via IP3R or RYR (not depicted) and is internalized in mitochondria through a voltage-dependent anion channel (VDAC) (not depicted), which is on the outer mitochondrial membrane, and through the mitochondria calcium uniporter (MCU), which is on the inner mitochondrial membrane [18]. Consequently, SEPN1 loss also impairs mitochondrial calcium and ATP levels by hampering the OXPHOS.

**Figure 3 cells-10-01116-f003:**
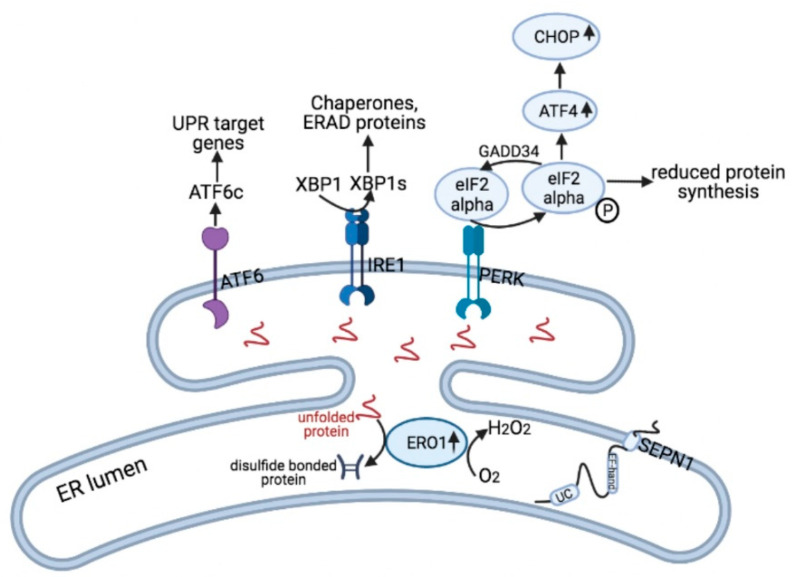
SEPN1 and ER stress. Unfolded proteins trigger ER stress and the consequent UPR by the activation of three sensors ATF6, IRE1 and PERK. Generally, this is a homeostatic response that restores ER homeostasis. After being proteolytically cleaved in the Golgi, ATF6 works as a transcription factor of UPR target genes. IRE1 promotes the splicing of XBP1 into XBP1s, which also works as a transcription factor for chaperones and ERAD proteins. PERK reduces protein synthesis by the phosphorylation of eif2 alpha and promotes the transcription of ATF4. The PERK pathway is also connected to pro-apoptotic signals via the transcription factor CHOP that, through the induction of GADD34, promotes recovery of the translation and, through ERO1, promotes the oxidative protein folding. However, ERO1 during oxidative protein folding also generates the oxidant H2O2, thus being involved in hyperoxidation. The expression of the type II membrane protein SEPN1 is regulated by ER stress and, through a redox reaction which involves the thioredoxin-like (CU) domain, counteracts ERO1-mediated hyperoxidation.

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
