# Peer review of "Calcium and Redox Liaison: A Key Role of Selenoprotein N in Skeletal Muscle"

_cells, 2021, doi:10.3390/cells10051116_

Round 1

Reviewer 1 Report

This is an interesting review that makes the link between molecular observations on one hand and pathologies and possible therapies on the other hand.

Some points however require clarifications:

(1) It is emphasized that SEPN1 is localized at MAMs and that this localisation is important for ATP production, which is of course highly significant. However, how this occurs is not clearly explained in the manuscript. Ca2+ transfer from the ER into mitochondria requires the presence of IP3R or RyR. The scheme (Fig. 2) is thus misleading and nothing about this is mentioned in the text. The possible involvement of IP3R is not mentioned, although expression of these receptors in skeletal muscles has been reported. 

If SEPN1  "only" acts by increasing the ER/mito contact sites, this effect should be clearly distinguished from its effect on SERCA activity.

(2)The authors seem to suggest that SEPN1 and RyR1 mutations have similar consequences. However, the Ca2+ fluxes mediated by these 2 channels are opposite. This should be clarified.

(3) The presence of a Ca2+ leak from the ER (recently reviewed by Lemos et al., BBA 2021) is not mentioned, although it is known that this leak is closely related to UPR.

(4) A scheme presenting the parts of the UPR pathway that are discussed in the review would be helpful. In the same line, I would advise the authors to speak about the PERK branch of the pathway, instead of the CHOP branch.

(5) If SEPN1 reduced activity can trigger ER stress via activation of CHOP, what about the other branches of UPR, which are equally triggered by Ca2+ depletion? The authors allude to this point at line 222, but it should be elaborated. 

(6) At several locations, the text is a bit hard to follow, mainly because some sentences are too long and because of the use of non defined abbreviations. Here below, I'm mentioning a few examples but I would advise a thorough re-reading of the text:

Lines 69-73. This sentence is too complicated.

Lines 136-137. In the parenthesis, what is the meaning of "of a synthetic peptide...EF-hand" ?

Lines 155-159. This sentence is too complicated.

Line 185. Is reference 30 appropriate?

Line 203. FDB muscle?

Lines 216-217. If I understand correctly, it should be "SEPN1-regulated ER stress" and not the opposite. 

Line 250. NAC?

Author Response

We thank the reviewer for their constructive critiques.

Please, find our point-by-point replies to the reviewers’ comments below: the comments are written in Arial bold type and our response in Italics.

(1) It is emphasized that SEPN1 is localized at MAMs and that this localisation is important for ATP production, which is of course highly significant. However, how this occurs is not clearly explained in the manuscript. Ca2+ transfer from the ER into mitochondria requires the presence of IP3R or RyR. The scheme (Fig. 2) is thus misleading and nothing about this is mentioned in the text. The possible involvement of IP3R is not mentioned, although expression of these receptors in skeletal muscles has been reported. 

Following the reviewer's critique we have modified the figure legend of fig.2  e also the text (line 115-117) including the description of IP3R and RyR.

(2)The authors seem to suggest that SEPN1 and RyR1 mutations have similar consequences. However, the Ca2+ fluxes mediated by these 2 channels are opposite. This should be clarified.

We have added few sentences to clarify this point  (lines 253-256)

(3) The presence of a Ca2+ leak from the ER (recently reviewed by Lemos et al., BBA 2021) is not mentioned, although it is known that this leak is closely related to UPR.

Although an important point, we don't believe that calcium leak is relevant in SEPN1-RM.

(4) A scheme presenting the parts of the UPR pathway that are discussed in the review would be helpful. In the same line, I would advise the authors to speak about the PERK branch of the pathway, instead of the CHOP branch.

(5) If SEPN1 reduced activity can trigger ER stress via activation of CHOP, what about the other branches of UPR, which are equally triggered by Ca2+ depletion? The authors allude to this point at line 222, but it should be elaborated. 

 To answer the reviewer's comments 4 and 5 we have added a new Figure 3 which depicts the different UPR branches discussed in the text.

(6) At several locations, the text is a bit hard to follow, mainly because some sentences are too long and because of the use of non defined abbreviations. Here below, I'm mentioning a few examples but I would advise a thorough re-reading of the text:

Thanks, all the text was carefully checked for typos and complicated sentences were rewritten.

Reviewer 2 Report

The present manuscript titled Calcium and redox liaison: a key role of SEPN1 in skeletal muscle.

Our appreciation of the role of endoplasmic reticulum (ER) stress pathways in both skeletal muscle homeostasis and the progression of muscle diseases is gaining momentum. Understanding the ER stress–related molecular pathways underlying physiologic and pathological phenotypes in healthy and diseased skeletal muscle should lead to novel therapeutic targets for muscle disease.

So, this review has two main sections titled Redox modulation of calcium handling in the ER and mitochondria and SEPN1–mediated calcium handling in the ER and mitochondria.

In pubmed we search “SEPN1 in skeletal muscle”, totally have 258 papers.

In pubmed search key words “Redox modulation of calcium handling in the ER and mitochondria”, totally have 1271 papers, The author only cited 11 papers in this section.

In pubmed search key words “SEPN1–mediated calcium handling in the ER and mitochondria” have 23 papers. The author only cited 9 papers in this section.

I strong suggest the author cite more reference, such as

Defective endoplasmic reticulum-mitochondria contacts and bioenergetics in SEPN1-related myopathy.Cell Death Differ. 2021 Jan; 28(1): 123–138. Published online 2020 Jul 13. doi: 10.1038/s41418-020-0587-z

And in the section SEPN1 at ER-to-mitochondria contact sites, the author also not discuss more details on the ER-to-mitochondria contact sites.

In the section Conclusion and therapeutic perspectives of core diseases, I strong suggest the author should discuss more about the SEPN1 in ER-to-mitochondria contact sites, should show the ideas how to identify other drugs targeted in ER-to-mitochondria contact sites.

Author Response

We thank the reviewer for their constructive critiques.

Please, find our point-by-point replies to the reviewers’ comments below: the comments are written in Arial bold type and our response in Italics.

In pubmed search key words “Redox modulation of calcium handling in the ER and mitochondria”, totally have 1271 papers, The author only cited 11 papers in this section.

In pubmed search key words “SEPN1–mediated calcium handling in the ER and mitochondria” have 23 papers. The author only cited 9 papers in this section.

I strong suggest the author cite more reference, such as

Defective endoplasmic reticulum-mitochondria contacts and bioenergetics in SEPN1-related myopathy.Cell Death Differ. 2021 Jan; 28(1): 123–138. Published online 2020 Jul 13. doi: 10.1038/s41418-020-0587-z

We have added 6 more references, as suggested by the reviewer. Furthermore the paper doi: 10.1038/s41418-020-0587-z was already cited.

And in the section SEPN1 at ER-to-mitochondria contact sites, the author also not discuss more details on the ER-to-mitochondria contact sites.

We have added more details on MAMs (lines 112-116)

In the section Conclusion and therapeutic perspectives of core diseases, I strong suggest the author should discuss more about the SEPN1 in ER-to-mitochondria contact sites, should show the ideas how to identify other drugs targeted in ER-to-mitochondria contact sites.

We have added: "At the moment we don’t know whether these alterations of MAMs in SEPN1-deficient models are due to a structural function of SEPN1, i.e to hold together ER and mitochondria, or are a consequence of other SEPN1-dependent alterations, i.e. ER stress." (lines 177-180)